# Exploring the spatial spillover effects of Yangtze River Delta ports on urban economic growth

Jian Hou[1], Juming Shi[1]*, Liangyu Chen[2], Zhouping Zhang[1], Edwin Kuang[3]

**1** School of Economics and Management, Shanghai Maritime University, Shanghai, China, **2** School of Management, Gachon University, Gyeonggi-do, South Korea, **3** School of Business, University of California, Riverside, California, United States of America

* shijuming@stu.shmtu.edu.cn

**Data Availability Statement:** All relevant data are within the paper and its Supporting Information files.

**Funding:** This study was funded by The National Social Science Fund Project (Grant No.22BTJ018).

## Abstract

The Yangtze River Delta (YRD) ports are pivotal in shaping the Yangtze River Economic Belt and advancing urban economies across China. This article utilizes panel data from 20 cities with ports in the YRD area, spanning from 2011 to 2020, using the spatial Durbin model to explore how these ports influence urban economic growth. The findings indicate that: (1) The YRD ports significantly contribute to economic growth in both the port cities and their surrounding areas, with the indirect impact on neighboring cities being more substantial than the direct effect on the cities themselves; (2) The beneficial spillover effects of the YRD ports on the economic growth of nearby cities vary in intensity over different spatial ranges, marked by distinct boundary effects and geographical attenuation. The influence extends up to approximately 110km; (3) Within the various elements impacting the economic growth of cities in the YRD, financial development prominently exhibits a threshold effect on urban economic growth; (4) Upon analyzing heterogeneity, inland and coastal port cities manifest divergent spillover effects, with inland port cities predominantly exerting a positive spillover on adjacent regions. Accordingly, in order to eventually achieve the shared prosperity of the region's economy, it is recommended that a strong top-level design be established and that efforts be made to transform the YRD region into a core region a diffusion and driving effect.

## 1. Introduction

Since the end of the last century, the notable acceleration of economic globalization has promoted international division of labor among nations or regions, a process largely reliant on an efficient and cost-effective maritime transport system. Maritime transportation, renowned for its low costs and extensive transport capacity, facilitates approximately 80% of the global trade volume [1]. This mode of transportation's pivotal role not only underscores the integral position of ports in the global trade network but also cements their status as fundamental infrastructure within shipping and contemporary logistics networks.

In the realm of international trade, ports are regarded as the linchpin of logistics and transportation. The presence of a significant port often determines whether a city is considered a

The person in charge of this project at the National Social Science Foundation of China is Prof. Guoliang Fan. The funder had no role in study design, data collection and analysis, decision to publish, or preparation of the manuscript.

**Competing interests:** The authors have declared that no competing interests exist.

critical central city within a country. Cities endowed with major ports command considerable influence over national economic trade, cultural exchanges, and openness to the international community. Such port cities boast sophisticated logistics networks, cutting-edge transportation infrastructure, robust industrial support, and abundant human resources. These attributes render them key players in international trade and pivotal engines driving national development [2]. Additionally, large port cities often attract numerous international businesses and investments, serving as hubs for innovation and technological development, and possessing distinct advantages in fueling urban and regional economic growth and social development [3]. Consequently, the scale and capacity of a city's port are often reflective of its stature and impact on both a national and international level.

The Yangtze River Delta (YRD) region, encompassing Shanghai and select areas of Jiangsu and Zhejiang provinces, is celebrated for its flourishing manufacturing and high-tech industries, alongside a vigorous financial services sector. This makes it one of China's most advanced and dynamic economic regions, serving as a critical growth pole. Distinguished by its abundant water resources and favorable water transportation conditions, the YRD's port infrastructure plays a crucial role. Amidst the backdrop of globalization and the international division of labor, the YRD's sophisticated port facilities have positioned it as an essential node within both China's and the global trade networks. These ports not only serve as conduits linking domestic and international markets but also act as vital engines propelling regional economic collaborative development, exerting a significant influence on the global trade landscape. The Chinese government's strategic focus on and enhancement of the YRD integration strategy, particularly through the enactment of the "14th Five-Year Plan for the Integrated Development of the Yangtze River Delta" in 2021, accentuates the pivotal role of these ports in regional economic development. Thus, a comprehensive exploration of the economic impacts of YRD port cities is paramount, enriching the theoretical foundations of economics, geography, and urban planning, while providing empirical support for policy-making, fostering the economic proliferation of port cities, and ensuring the efficient functioning of ports.

YRD ports are not merely crucial gateways for China's integration into the global sphere but also primary catalysts for the swift economic development and integration of the region. By facilitating the efficient allocation and circulation of resources, these ports have knit tight economic ties and a robust logistics network, acting as essential hubs for both the aggregation and distribution of goods, as well as pivotal junctions for the flow of information, capital, and talent. As integral components of the industrial chain, YRD ports have forged close upstream and downstream relationships with adjacent manufacturing, service, and various industries, attracting a significant aggregation of production and service enterprises through their logistical and locational benefits. These clusters, through the sharing of resources, technology, and market information, further accelerate the integrated and collaborative development of the regional economy. Studying the spatial spillover effects of YRD ports on urban economic growth offers policymakers a scientific basis for port and regional collaborative development strategies, aiding in the formulation of precise policies aimed at promoting the coordinated and balanced advancement of ports and the regional economy, thereby possessing significant practical value.

In the literature review section, this paper organizes related literature, and compared to current research, presents the marginal contributions of this article: (1) This paper focuses its research scope on the YRD region, not only enabling a deeper analysis of the region's uniqueness but also proposing more actionable countermeasure suggestions. (2) This paper conducts ordinary regression analysis on the economic development levels of various cities and their geographical distances from Shanghai, preliminarily identifying the spatial connections between port cities. (3) It analyzes how the spatial distance between port cities affects the

spillover effects of urban economic growth and determines the geographical decay boundaries of spillover effects. (4) Utilizing a spatial Durbin model with heterogeneity coefficients, it studies the spatial spillover effects of each port city on economic growth. (5) Through heterogeneity analysis, it explores the differences in spillover effects exhibited by coastal port cities and inland port cities in urban economic growth. (6) Using a threshold effect model, it investigates the nonlinear role of financial development in the impact of ports on urban economic growth.

## 2. Literature review

### 2.1. Influences of ports on the regional economy

Extensive research confirms that ports enhance the flow of goods and capital, diminish transaction costs, and widen market connectivity [4, 5], yielding significant economic and social benefits for local cities. Specifically, the development of ports can directly influence the growth of related industries, such as loading and unloading, logistics, warehousing, and equipment manufacturing, among other port-related industries. Meanwhile, in cities with more developed financial sectors, the expansion of ports can also lead to the rise of port service industries such as insurance, financial, and commercial services [3, 6].

In reviewing the past studies on the relationship between ports and regional economic growth, this paper finds that scholars have utilized a variety of methods to explore the impact of ports on the hinterland economy. These include traditional input-output analysis [7], grey relational analysis [8], and structural equation modeling [9]. These methods uncover the intricate relationship between port activities and regional economies from diverse angles. However, such methods have limitations and do not consider the spatial agglomeration characteristics of both ports and cities. Nonetheless, as key nodes of urban economic activities, the impact of ports on regional economies does not occur in isolation. Ports may generate spillover effects in the surrounding areas through trade dynamics and market adaptation mechanisms, which can have both positive and negative impacts on the economies of adjacent areas [10]. Furthermore, since port cities are more likely to gain comparative economic advantages, they often become the epicenters of economic agglomeration [11]. Under the influence of agglomeration forces, urban economies may self-reinforce, thereby gaining momentum for long-term economic growth [9]. This agglomeration can self-reinforce urban economies, propelling long-term growth and extending spillover effects to surrounding regions.

In recent years, the topic of whether ports can generate economic spillover effects on surrounding areas has gradually sparked heated discussions, contributing to a rich literature. With the growth of global trade volume, the flow of cargo handled by ports has significantly increased, necessitating the improvement of corresponding transportation infrastructure. Consequently, ports can extend their economic impact to surrounding areas through a widespread transportation network [12]. Clearly, the spillover effects of ports are complex and can have either positive or negative impacts on surrounding areas, leading to no unanimous conclusion. For instance, Bottasso et al., in their study of the impact of Western European ports on regional economies, found using spatial econometric models that an increase in port throughput also raises the local GDP, and has a positive spillover effect on the GDP of neighboring regions, indicating that port activities generate positive externalities [10]. However, negative impacts can also arise. For example, Cohen and Monaco using data from California, USA, found that investment in port infrastructure could promote the development of the state's retail industry but produced a negative spillover effect on surrounding areas, creating a "siphoning effect" [13].

As a country with an extensive coastline, the construction and development of ports have always been an integral part of China's economic strategy. Studies on the impact of Chinese

ports on regional economic development are widespread, focusing broadly on industrial agglomeration, employment and social development, and regional economic growth and integration, among others. In terms of industrial agglomeration, Xu and Wang found that ports have played a promotional role in the logistics industry agglomeration in the YRD region, and this type of industry agglomeration further promotes regional economic growth [14]. Regarding employment, Wang and Wang believe that port investment stimulates potential demand in upstream and downstream industries, directly and indirectly increasing the output of other industries to promote employment [15]. In terms of regional economic growth, Deng et al. using the spatial Durbin model to study 46 coastal ports in China, found that ports have a significant economic spillover effect on surrounding cities [16].

In the process of reviewing the existing literature, it was found that current research mainly focuses on integrating various port clusters for study, rarely conducting separate analyses. However, considering China's major port clusters include the Bohai Rim, Yangtze River Delta, Pearl River Delta, Southeast Coast, and Southwest Coast regions, each port cluster's development level is not the same, and the distances between each port cluster are further compared to the distances within the port clusters themselves. If studied collectively, it might underestimate or overestimate the role of individual port clusters in promoting economic growth in surrounding cities. Therefore, this paper considers focusing the research scope within the YRD port area, making the research conclusions clearer and more comprehensive, and thus the countermeasure suggestions more actionable.

While numerous studies have addressed the spillover effects of ports on regional economies, few have delved into the specifics of these effects' boundaries. Recognizing that spillover effects diminish with increased spatial distance, and that surrounding cities experience these effects differently, identifying the spatial limits of these effects is crucial. Such insight would enable higher-level governments to craft strategic plans that maximize the beneficial impacts of ports, fostering economic integration and balanced development across regions. Additionally, the variation in development levels between ports, along with the distinct functions served by inland versus coastal ports, results in varied spillover effects. Yet, there appears to be a gap in existing research regarding a detailed analysis of these distinctions. Therefore, this paper seeks to fill this gap by adopting a heterogeneity-focused approach to examine the distinct spillover effects generated by individual ports, distinguishing between inland and coastal ports. This perspective aims to provide a more comprehensive understanding of how different types of ports contribute to regional economic dynamics.

## 2.2. The role of financial development in the impact of ports on regional economies

Financial development holds a significant link to the economic dynamics of port areas, serving as a crucial component of the socio-economic wealth in these regions [17]. In the studies related to the relationship between ports and regional economics, some scholars have affirmed that financial development can promote the economic prosperity of port areas. For example, Hesse and Rodrigue highlighted that financial development underpins the construction and operational aspects of logistics infrastructure and goods distribution, further amplifying economic benefits across regions through the transportation network [18]. However, the influence of financial development on the relationship between ports and regional economic growth is not linear; its effects may become ineffective or even produce negative impacts. For instance, Musso et al. examined the contrasting impacts of public and private investments on regional economies linked to ports, noting that private investments might neglect external positives, while public investments could result in the inefficient allocation of financial resources,

failing to bolster port-related industry development effectively [19]. Moreover, the relationship between financial development and the regional economy concerning ports is also affected by regional economic structures, policy environments, legal frameworks, and market maturity [20].

Reflecting on the existing literature, found that scholars tend to explore the interaction between financial development and port economies without considering the role played by different levels of financial development in influencing the impact of ports on regional economies. Therefore, how to effectively manage and control financial development to maximize its positive effects in the economic influence of ports on their regions is the focus of this study. To achieve this, the paper employs the threshold effect model to investigate how different financial development stages modify the capacity of ports to spur regional economic growth.

## 3. Mechanism analysis and research hypotheses

The YRD port cluster is among China's paramount port conglomerates, exerting substantial influence on the region's economy through its geographical advantages. Initially, the port manifests pronounced industrial agglomeration characteristics, adept at engendering agglomeration effects [21]. As focal points for the convergence of goods, capital, information, and technology, ports leverage a fundamental advantage as platforms for the integration of production factors. To capitalize fully on this advantage, related enterprises often cluster in the vicinity, bolstering cooperation and thereby continuously augmenting industrial scale. This process results in economies of scale, diminishing transportation, production, and transaction information costs, and consequently attracting more enterprises, further expanding industrial scale, and better driving economic growth. Additionally, the port's locational, infrastructural, and logistical service advantages encourage the agglomeration of enterprises from related industries within port areas. This promotes the concentration of labor, production materials, technology, and capital, facilitating the seamless flow of production factors and thereby propelling economic growth.

Secondly, as a core link in the supply chain, ports can significantly strengthen the synergistic effect between upstream and downstream enterprises by improving the efficiency of cargo circulation and reducing transaction costs [22]. This synergy enhances the mobility of production factors, accelerates the speed of social circulation, and thus improves the response speed and market adaptability of the entire supply chain. Specifically, efficient supply chain synergy bolsters the responsiveness of upstream industries to market fluctuations and furnishes downstream industries with effective raw material supply and product distribution channels. This not only promotes closer economic ties between industries but also enhances the overall efficiency of the regional economy by improving supply chain efficiency.

Lastly, ports also promote trade facilitation, reduce import and export costs, attract more international trade and foreign investment, and increase the openness and attractiveness of the regional economy [18]. Infrastructure improvements, such as the construction of port facilities, roads, and railways, not only directly create jobs but also attract more private and public investment, promoting capital formation and accumulation, providing momentum for the sustained growth of the regional economy. Based on the above analysis, this paper proposes Hypothesis 1.

**Hypothesis 1:** Yangtze River Delta ports can promote local city economic growth.

Ports, serving as convergence points of water and land transportation and the linchpins of regional economic and trade activities, possess distinct network characteristics [23]. By leveraging their network nature, ports can enhance economic and trade exchanges between

regions, accelerate the aggregation and diffusion of logistics and economic activities, and promote the flow of production factors and goods between regions, thereby potentially generating positive or negative spatial spillover effects on regional economic growth [24]. On one hand, ports can generate positive spatial spillover effects on the economic growth of surrounding areas by facilitating the smooth flow of production factors and products between regions. Furthermore, as port infrastructure continuously improves, its linking function gradually strengthens, increasing the economic connections between adjacent areas. The increasingly developed transportation conditions enhance the positive spatial spillover effects on the economic growth of surrounding areas. On the other hand, a well-developed port infrastructure network will direct logistic elements towards economically developed regions, thereby maintaining their advantageous position. Before the peripheral lagging regions can effectively enhance their absorption and development capabilities, they are unable to benefit from the positive spillover effects generated by the port logistics industry of the developed regions. Moreover, if the diffusion effects are less than the agglomeration effects, it could result in the lagging regions experiencing negative spillover effects from the developed areas, thereby exacerbating the imbalance in regional economic development.

The spillover effects of ports on the economies of adjacent regions are determined by multiple factors, including the level of infrastructure, the economic development of the region where the port is located, connectivity and radiating capability, as well as the adaptability, development strategies, and learning imitation effects of surrounding areas [25]. Advanced infrastructure and efficient logistics systems allow certain ports to promote inter-regional economic activities, attract international trade and investment, bringing positive economic spillover effects. However, if the region where the port is located is already economically developed, it may exacerbate regional economic imbalances, draw resources away from surrounding underdeveloped areas, and generate negative spillover effects. Moreover, the connectivity and radiating capability of a port determine its sphere of influence, while the development strategies and learning imitation effects of surrounding areas determine their ability to benefit from port development. Therefore, whether a port can generate positive spillover effects on surrounding areas or lead to negative impacts depends on the combined action of these factors. Finally, coastal ports mainly serve international trade, while inland ports are more oriented towards domestic trade. International trade often involves longer-distance goods transportation and more logistics links, meaning that the economic activities of coastal ports are more likely to spread globally, with relatively weak economic ties to surrounding areas. In contrast, inland ports support more domestic economic circulation, involving more domestic distribution and consumption, and therefore have stronger economic interaction and dependence with surrounding areas.

Based on the first law of geography "Everything is related to everything else, but near things are more related than distant things" [26], the impact of local cities on surrounding areas may be limited by spatial distance [27]. The speed and strength of economic spillovers are related to geographical distance, and the increase in spatial distance gradually weakens the intensity of interaction between these spatial units. As the distance increases, the economic spillover effects of ports gradually weaken. For example, increased transportation costs make it more expensive for cities far from ports to obtain raw materials and export products, reducing their competitiveness; the radiating effect of industrial agglomeration makes it difficult for cities far from ports to enjoy the direct spillover effects of port economic activities; the spread of information and technology is also hindered, affecting these cities' economic innovation and development speed. Therefore, the economic spillover effects of ports on surrounding cities are influenced by distance. Based on this analysis, this paper proposes Hypothesis 2, Hypothesis 3, and Hypothesis 4.

**Hypothesis 2:** Ports in the Yangtze River Delta will have a positive spillover effect on the economies of nearby regions.

**Hypothesis 3:** Different ports have varying economic spillover effects on cities, with coastal ports and inland ports having different impacts on surrounding areas.

**Hypothesis 4:** With the increase in distance between ports and cities, the spillover effect of the Yangtze River Delta ports exhibits decay and boundaries.

Financial development enhances the depth and efficiency of capital markets, promoting the effective accumulation and allocation of capital. This provides the necessary financial support for the construction of port infrastructure and the expansion of related industries [28], thereby enhancing the throughput capacity and operational efficiency of ports, which helps to drive regional economic growth. However, financial development can also exacerbate the misallocation of resources in ports and related industries. Considering that financial institutions might favor investing in projects with high short-term returns while neglecting the importance of long-term infrastructure construction and maintenance, this could affect the long-term competitiveness of ports and the stability of regional economies [19]. Additionally, financial intermediaries, in providing funding and risk management services, may experience economies of scale and scope. Below a certain scale, the marginal cost of financial intermediation is high, but beyond a critical point, an increase in scale can significantly reduce costs, playing a facilitating role in the process of ports promoting regional economic growth. However, when the scale of financial intermediation becomes too large, problems such as increased management complexity and exacerbated information asymmetry may lead to decreased efficiency, thereby affecting the role of ports in promoting regional economies to some extent. Based on this analysis, this paper proposes Hypothesis 5.

**Hypothesis 5:** In the impact of ports on urban economic growth, financial development plays a role and exhibits a threshold effect.

## 4. Research methods

### 4.1. Sample and data sources

Given the dense water network and close distribution of ports in the YRD, to ensure the empirical analysis results are persuasive, the selection of samples needs to be representative. This paper selects major ports located along the Yangtze River or its tributaries in Jiangsu Province, Zhejiang Province, and Shanghai City, using prefecture-level cities as units for the study. Coastal ports mainly include Shanghai Port, Lianyungang Port, Yancheng Port, Ningbo-Zhoushan Port, Jiaxing Port, Wenzhou Port, and Taizhou Port. Ports along the Yangtze River mainly consist of Nanjing Port, Suzhou Port, Wuxi Port, Taizhou Port, Yangzhou Port, Nantong Port, Changzhou Port, and Zhenjiang Port. The inland ports mainly cover Xuzhou Port, Huai'an Port, Hangzhou Port, Shaoxing Port, and Huzhou Port. The sample ports selected for this study total 20, covering the research period from 2011 to 2020. Data for the variables comes from the "China Port Yearbook" and the statistical yearbooks of each city.

### 4.2. Selection of variables

This paper investigates the impact of the YRD ports on city economic growth and spatial spillover effects, using the city's economic development level as the dependent variable, measured by per capita GDP and represented by PGDP. The core explanatory variable is the level of port development, measured by port throughput and represented by PT. The corresponding

**Table 1. Variable information.**

| Variable | Obs | Mean | Std. Dev. | Min | Max |
|---|---|---|---|---|---|
| lnPGDP | 200 | 11.343 | 0.364 | 10.383 | 12.016 |
| lnPT | 200 | 9.556 | 0.942 | 7.078 | 11.672 |
| lnSSL | 200 | 10.238 | 0.566 | 9.225 | 11.875 |
| lnCSL | 200 | 0.38 | 0.074 | 0.228 | 0.589 |
| INF | 200 | 0.018 | 0.007 | 0.006 | 0.051 |
| lnHUM | 200 | 2.391 | 0.878 | 0.928 | 4.481 |
| FIN | 200 | 1.090 | 1.344 | 0.099 | 7.799 |
| lnDIS | 200 | 5.085 | 1.261 | 0 | 6.228 |
| lnGDP | 200 | 17.829 | 0. 673 | 16.462 | 19.773 |

control variables are as follows: the number of hospital beds (units) is selected to measure the city's social security level and is represented by SSL; total retail sales of consumer goods (ten thousand yuan) to measure the level of urban residents' consumption and is represented by CSL; the sum of total postal and telecommunications services (ten thousand yuan) as a percentage of GDP to measure the level of informatization, represented by INF; the number of students enrolled in regular higher education institutions (ten thousand people) to measure the level of urban human capital, represented by HUM. In addition, the year-end balance of loans from financial institutions (billion yuan) is selected to measure the level of financial development, represented by FIN. Except for the level of informatization, which is a percentage, other indicators in this paper are processed in natural logarithms.

To test the robustness of the conclusions in this paper, we also consider conducting regression analysis using the annual gross domestic product of cities (denoted as GDP) as the dependent variable. Moreover, considering Shanghai, Tianjin, and Hong Kong are China's three major coastal ports, which play a significant role in promoting the economy of surrounding areas, and given the proximity of the YRD ports to Shanghai, this paper calculates the distance between the sample cities and Shanghai (denoted as DIS) using geographical coordinates. This distance is used as a core explanatory variable in the basic regression to preliminarily test spatial correlation. Detailed data are provided in Table 1.

### 4.3. Basic model

Before formally employing spatial econometric models, it is necessary to observe the basic model results, that is, to test whether the YRD ports have a positive effect on urban economic growth. Therefore, this paper sets the basic panel model as:

$$\ln PGDP_{it} = \beta \ln PT_{it} + \alpha_k X_{it} + \mu_i + \lambda_t + \varepsilon_{it} \tag{1}$$

In model (1), $i$ signifies cities, and $t$ denotes years. $\mu_i$ represents individual fixed effects, $\lambda_t$ represents time fixed effects, and $\varepsilon_{it}$ represents the random error term. $PGDP_{it}$ stands for the city's economic development level, $PT_{it}$ is the core explanatory variable representing the port development level. $X_{it}$ refers to control variables, including the level of social security, the level of residents' consumption, the level of informatization, and the level of human capital.

### 4.4. Spatial econometric model

New Economic Geography suggests that there is a certain correlation between different geographic locations, which means that when studying the impact of the YRD ports on urban economic growth, it is necessary to consider this spatial interaction. Therefore, spatial

econometric models are employed for analysis. Common spatial econometric models include the Spatial Lag Model (SLM), Spatial Error Model (SEM), and Spatial Durbin Model (SDM). Considering this paper aims to study the spatial spillover effects of ports on urban economic growth, with urban economy as the dependent variable and ports as the independent variable. Simply put, SLM and SEM models analyze the spillover effects through the spatial lag of the dependent variable, meaning they look at how urban economies influence each other but cannot determine what causes these spillover effects. The SDM model, on the other hand, not only considers the spatial lag effects of the dependent variable (how the dependent variables of neighboring areas affect the dependent variable of the local area) but also the spatial lag effects of the independent variables (how the independent variables of neighboring areas affect the dependent variable of the local area) and possible spatially correlated errors, making it an extension or combination of the SLM and SEM models. Since the SDM model considers both the spatial lags of independent and dependent variables, it provides a more comprehensive perspective for analyzing spatial spillover effects, allowing for a more precise distinction and identification of different types of spatial effects, such as direct effects (local ports on local economy) and indirect effects (local ports on the economy of neighboring areas). Therefore, this paper employs the SDM model for the study, with the related models as follows:

$$\ln PGDP_{it} = \alpha_0 + \rho W \ln PGDP_{it} + \alpha_1 \ln PT_{it} + \beta_1 W \ln PT_{it} + \alpha_k X_{it} + \beta_k W X_{it} + \delta_i + \mu_t + \boldsymbol{\varepsilon}_{it} \quad (2)$$

In the above formula (2), $i$ represents the 20 the YRD port cities; $t$ represents the research period of 10 years, $\rho$ is the spatial autoregressive coefficient of the dependent variable, W indicating the spatial weight matrix, $\alpha_1$ is the regression coefficient of the core explanatory variable, $\beta_1$ is the spatial lag coefficient of the core explanatory variable, X represents various control variables, $\alpha_k$ is the regression coefficient of control variables, $\beta_k$ is the spatial lag coefficient of control variables, $\mu_i$, $\lambda_t$, and $\varepsilon_{it}$ have the same meaning as in formula (1).

## 4.5. Spatial weight matrix

The main difference between spatial econometric models and traditional econometric models lies in that the former considers the spatial interactions and individual differences among the subjects under study, that is, spatial effects. Spatial effects in econometric models are primarily reflected through the introduction of a spatial weight matrix. Therefore, choosing an appropriate spatial weight matrix is quite important for the construction of spatial econometric models.

**4.5.1. 0–1 matrix.** If the two observations under study are adjacent in geographical space, meaning they share a common boundary, then the weight is 1; otherwise, the weight is 0. This is based on the queen contiguity criterion. The expression for the adjacency matrix is as follows:

$$W_{0-1} = \begin{cases} 1, & \text{Region } i \text{ shares a border with region } j \\ 0, & \text{Region } i \text{ does not share a border with to region } j \end{cases} \quad \text{Here}, i \neq j \quad (3)$$

**4.5.2. Geographic distance matrix.** The geographic distance matrix determines the spatial correlation between the areas where the subjects of study are located based on the magnitude of their geographical distance. This paper calculates the geographic distances between cities using the latitude and longitude coordinates from the national geographic information system and constructs the weight matrix using the reciprocal of geographical distance. The

expression for the geographic distance weight matrix is as follows:

$$W_1 = \begin{cases} \dfrac{1}{d_{ij}}, & i \neq j \\ 0, & i = j \end{cases} \tag{4}$$

In expression (4), $\frac{1}{d_{ij}}$ is the inverse of the distance based on latitude and longitude between region $i$ and region $j$. Based on this, we can further construct the inverse distance squared weight matrix $W_2$, when $i \neq j$, $W_{2_{ij}} = \frac{1}{d_{ij}^2}$; when $i = j$, $W_{2_{ij}} = 0$.

**4.5.3. Financial development distance matrix.** This paper adopts the method of constructing an economic distance matrix by Shao et al. to establish the financial development distance matrix W3 [29], as specified in formula (5):

$$W_{3_{ij}} = \frac{1}{\text{abs}(\text{perFIN}_i - \text{perFIN}_j)d_{ij}} \tag{5}$$

where $\text{perFIN}_i$ and $\text{perFIN}_j$ represent the average financial development in regions $i$ and $j$, and $d_{ij}$ denotes the geographical distance between them. The 'abs' stands for the absolute value function, which ensures that the weight values are non-negative. The absolute magnitude of the disparity between $\text{perFIN}_i$ and $\text{perFIN}_j$ indicates the degree of difference in the financial development between regions $i$ and $j$. This means that regions with smaller differences have higher weight values, while regions with larger differences have lower weight values.

## 4.6. Threshold test for financial development in the effects of the YRD ports on urban economic growth

This paper adopts the method developed by Hansen to construct the model with financial development as the threshold variable [30], as shown in formula (6).

$$\ln \text{PGDP}_{it} = \alpha_0 + \alpha_1 \ln \text{PT}_{it} \cdot I(\text{FIN}_{it} \leq \gamma_1) + \alpha_2 \ln \text{PT}_{it} \cdot I(\gamma_1 \leq \text{FIN}_{it} \leq \gamma_2) + \cdots +$$

$$\alpha_n \ln \text{PT}_{it} \cdot I(\gamma_{n-1} \leq \text{FIN}_{it} \leq \gamma_n) + \varphi_k X_{it} + \mu_i + \lambda_t + \varepsilon_{it} \tag{6}$$

The FIN is the threshold variable, $\gamma_1 \cdots \gamma_n$ are the threshold values, $I(\cdot)$ is the indicator function that takes the value of 1 when the condition within the brackets is met, otherwise it is 0, $\alpha_0$ is the constant term, $\alpha_1 \cdots \alpha_n$ are the corresponding regression coefficients, and the meanings of the other letters are the same as in formula (1).

## 5. Empirical analysis

### 5.1. Basic regression

To examine the impact of the YRD ports on urban economic growth and the spatial correlation between the two, this paper conducts ordinary panel regressions of the port development level PT and the distance DIS between each city and Shanghai against the level of urban economic development. This preliminary test aims to explore whether there is a connection between ports and urban economies and whether it is logical to apply a spatial econometric model. Detailed results are provided in Table 2.

From the first column, port development (PT) has a positive effect on urban economic development (PGPD), indicating that ports can promote urban economic growth. From the second column, it is known that there is a significant negative correlation between the distance (DIS) of each sample city from Shanghai and the level of urban economic development

**Table 2. Basic regression.**

| lnPGDP | Value | lnPGDP | Value |
|---|---|---|---|
| lnPT | 0.1761*** | lnDIS | -0.0587*** |
| | (3.22) | | (-2.67) |
| lnSSL | 0.5755*** | lnSSL | 0.0349 |
| | (6.65) | | (0.49) |
| lnCSL | 1.0261*** | lnCSL | -0.0282 |
| | (3.01) | | (-0.08) |
| INF | -2.6292 | INF | -5.8709 |
| | (-1.03) | | (-1.57) |
| lnHUM | 0.8118*** | lnHUM | 0.1883*** |
| | (6.09) | | (4.95) |
| Constant | 1.4845** | Constant | 10.9509*** |
| | (2.15) | | (15.68) |
| R-squared | 0.699 | R-squared | 0.338 |
| Observations | 200 | Observations | 200 |

Attention

***, **, and * correspond to noteworthiness at the 1%, 5%, and 10% levels, respectively, throughout this paper.

(PGPD). This preliminarily reveals the potential spatial correlation between cities, providing a rationale for the introduction of spatial econometric models in the subsequent text.

## 5.2. Spatial Durbin model

**5.2.1. Global Moran's index.** This paper calculates the global Moran's I index to measure the spatial dependency of the dependent variable PGDP and the core explanatory variable PT, choosing the geographic distance matrix. According to Table 3, from 2011 to 2020, the global Moran's I index of PGDP and PT are all positive values and have passed the significance tests at 1% and 10% levels for all the years respectively. This indicates that there is a significant positive spatial correlation for both PGDP and PT in the cities of the YRD region. Therefore, spatial factors should not be overlooked when researching the issue of how the YRD ports promote urban economic growth.

**5.2.2. Model testing.** The Moran test indicates that there is a spatial effect between ports and urban economic growth, but relying solely on this to construct a spatial econometric

**Table 3. Moran's I Index values.**

| lnPGDP | | | | lnPT | | | |
|---|---|---|---|---|---|---|---|
| Year | Moran's I | Z-Value | P-Value | Year | Moran's I | Z-Value | P-Value |
| 2011 | 0.281 | 4.005 | 0.000 | 2011 | 0.178 | 2.083 | 0.019 |
| 2012 | 0.264 | 3.793 | 0.000 | 2012 | 0.165 | 1.974 | 0.024 |
| 2013 | 0.257 | 3.711 | 0.000 | 2013 | 0.139 | 1.749 | 0.040 |
| 2014 | 0.252 | 3.644 | 0.000 | 2014 | 0.142 | 1.758 | 0.039 |
| 2015 | 0.239 | 3.492 | 0.000 | 2015 | 0.114 | 1.522 | 0.064 |
| 2016 | 0.234 | 3.424 | 0.000 | 2016 | 0.094 | 1.320 | 0.093 |
| 2017 | 0.232 | 3.404 | 0.000 | 2017 | 0.138 | 1.730 | 0.042 |
| 2018 | 0.233 | 3.413 | 0.000 | 2018 | 0.134 | 1.669 | 0.048 |
| 2019 | 0.238 | 3.476 | 0.000 | 2019 | 0.161 | 1.875 | 0.030 |
| 2020 | 0.237 | 3.461 | 0.000 | 2020 | 0.178 | 2.023 | 0.022 |

**Table 4. Model testing.**

| Test Indicators | Z-Value | P-Value |
|---|---|---|
| Moran's I | 12.894 | 0.000 |
| LM—Lag | 74.897 | 0.000 |
| RLM—Lag | 4.038 | 0.044 |
| LM-Error | 129.941 | 0.000 |
| RLM-Error | 59.083 | 0.000 |
| LR—SAR | 30.48 | 0.000 |
| LR—SEM | 59.74 | 0.000 |
| Wald—SAR | 37.36 | 0.000 |
| Wald—SEM | 42.49 | 0.000 |
| Hausman | 29.61 | 0.002 |

model is not reasonable; other tests are also needed. Following the earlier discussion, using the SDM model for further research is more in line with the theme of this paper, but this requires a series of tests. First, the LM test is used to determine if SEM or SLM models can be applied; if the test fails, spatial econometric methods cannot be used. If the test is successful, then its robustness needs to be tested; if Robust LM test fails, only SEM or SLM models can be adopted. If the test is successful, then LR or Wald tests are used to decide if the SDM model can be applied. The logic behind these two tests is the same, to test whether the SDM can be simplified to SEM or SLM models. If the LR or Wald tests are passed, it proves that the SDM model is stable, and thus can be used for spatial econometric analysis.

Initially, the LM test is used to assess the applicability of SLM and SEM models, as depicted in Table 4. The test results show that under the SLM and SEM models, the LM and RLM test statistics passed the 1% and 5% significance levels, respectively, indicating that both the error term and the dependent variable PGDP exhibit significant spatial effects. Consequently, both SLM and SEM models are viable options. However, given the SDM model's capability to incorporate spatial effects of both independent and dependent variables, it is deemed superior. This necessitates proceeding with LR and Wald tests to evaluate whether the SDM model can be reduced to the SEM or SLM models. The outcomes presented in Table 4 reveal that the LR and Wald test statistics for the SEM and SLM models are significant at the 1% level, leading to the rejection of the null hypothesis that the SDM model can be simplified to the SLM or SEM models. In essence, the SDM remains robust and unaffected by potential degradation to SLM or SEM models during analysis. Thus, this study selects the SDM model for empirical analysis.

Finally, we use the Hausman test to determine whether to use fixed effects or random effects. From the Hausman test results in Table 4, the Hausman test statistic is 29.61, and the null hypothesis is rejected at the 1% significance level, indicating that the fixed effects SDM model should be chosen over the random effects SDM model. The Log-likelihood values were also compared, with the double fixed effects model having the highest value of 403.5533. Taking everything into consideration, this study selects the SDM model with both individual and time fixed effects.

**5.2.3. Regression results.** This paper constructs the SDM model using a geographic distance matrix and conducts regression analysis with STATA software, with results presented in Table 5. The spatial autoregressive coefficient of PGDP, ρ is 0.4047 and significant at the 5% level, indicating the presence of spatial connections, i.e., neighboring cities have a positive impact on the economy of the YRD port cities; moreover, the coefficient of lnPT is 0.098 and significant at the 1% level, suggesting that ports can promote economic growth in local cities; simultaneously, the spatial lag coefficient of lnPT is 0.9162 and significant at the 1% level,

**Table 5. SDM regression results.**

| Variable | Direct | Variable | Indirect |
|---|---|---|---|
| lnPT | 0.0980*** | W×lnPT | 0.9162*** |
| | (5.88) | | (6.97) |
| lnSSL | -0.1542*** | W×lnSSL | -1.2888*** |
| | (-4.38) | | (-7.25) |
| lnCSL | 0.0396 | W×lnCSL | 0.2155 |
| | (0.34) | | (0.24) |
| INF | -2.2222** | W×INF | -18.4516** |
| | (-2.38) | | (-2.15) |
| lnHUM | -0.0591 | W×lnHUM | -0.4317 |
| | (-1.25) | | (-1.26) |
| $\rho$ | 0.4047** | | |
| | (2.55) | | |
| Observations | 200 | | |
| Log-likelihood | 403.5533 | | |
| Number of ID | 20 | | |

indicating that ports also have a driving effect on the economy of nearby cities. Possible reasons: Ports serve as crucial transportation hubs connecting cities to external markets, not only fostering the prosperity of local commodity trade markets and expanding foreign trade but also accelerating the circulation of funds, technology, and information through goods transportation. This promotes economic interconnection and integration between regions, thereby generating positive spillover effects on the economies of surrounding cities, preliminarily verifying the establishment of Hypothesis 1 and Hypothesis 2.

**5.2.4. Effects decomposition.** This model incorporates the spatial lag of the core explanatory variable PT, and its estimated results may not directly reflect the marginal effects of the PT, making it difficult to accurately measure the impact of the YRD ports on urban economic growth. Therefore, it is necessary to decompose the spatial spillover effect into direct and indirect effects. Here, by adopting the methods of Elhorst (2014) and Lesage (2009) [31, 32], the coefficients are decomposed. This decomposition is achieved by calculating the partial derivative matrix of the dependent variable Y (PGDP) with respect to the independent variable X (PT), as shown in formula (7)

$$
\begin{bmatrix}
\dfrac{\partial Y_1}{\partial X_1} & \cdots & \dfrac{\partial Y_1}{\partial X_n} \\
\cdots & \cdots & \cdots \\
\dfrac{\partial Y_n}{\partial X_1} & \cdots & \dfrac{\partial Y_n}{\partial X_1}
\end{bmatrix}
= (I_n - \rho W)^{-1}
\begin{bmatrix}
\beta_k & \cdots & W_{1N}\phi_k \\
\cdots & \cdots & \cdots \\
W_{N1}\theta_k & \cdots & \beta_k
\end{bmatrix}
\tag{7}
$$

Each component on the right side of the equation signifies the partial derivative within the matrix of the Y of region *i* about the X of region *j*; In is an n×n identity matrix; The connotations of the $\rho$ and W are congruent with those delineated in Eq (2). $\beta_k$ represents the direct effect of the PT, that is, the average total change in the local city's economy caused by all port changes. Its economic meaning is the impact of ports on the local city economy; the off-diagonal values on the right side of the matrix represent the indirect effects, that is, the average total change in the economy of adjacent areas caused by all port changes. Its economic meaning is the impact of ports on the economy of neighboring area cities.

**Table 6. Spillover decomposition.**

| Variable | Direct | Indirect | Total |
|---|---|---|---|
| lnPT | 0.1428*** | 1.6733*** | 1.8161*** |
|  | (4.35) | (3.23) | (3.32) |
| lnSSL | -0.2173*** | -2.3370*** | -2.5543*** |
|  | (-4.70) | (-3.39) | (-3.54) |
| lnCSL | 0.0661 | 0.5002 | 0.5663 |
|  | (0.47) | (0.30) | (0.32) |
| INF | -3.1065** | -32.9622* | -36.0688* |
|  | (-2.27) | (-1.76) | (-1.81) |
| lnHUM | -0.0800 | -0.7735 | -0.8535 |
|  | (-1.41) | (-1.16) | (-1.21) |

As shown in Table 6, direct effect: for every 1% increase in port throughput, the local city's GDP per capita grows by 0.1428%. Indirect effect: for every 1% increase in port throughput, it contributes to a 1.6733% increase in the GDP per capita of surrounding cities. Comparing the direct and indirect effects, ports indeed promote the economic growth of cities in the YRD region, not only enhancing the local city's economic growth but also positively influencing the economies of surrounding cities. Moreover, the spillover effect coefficient is 1.6733, greater than the direct effect coefficient of 0.1428, indicating that a 1% increase in local port through-put leads to a 1.5305% greater increase in the GDP per capita of surrounding cities than in the port city itself. This demonstrates that compared to the impact of ports on the economic growth of their host cities, their role in driving the economic growth of neighboring cities is more significant. In summary, this supports Hypothesis 1 and Hypothesis 2.

**5.2.5. Spatial decay.** From the above analysis, it is known that ports have a positive spatial spillover effect, but the conclusion does not consider the differences in city distances. "The First Law of Geography" suggests that the spatial dependence of economic activities tends to decay with an increase in geographic distance. Does the spatial spillover effect of ports on urban economies also exhibit this characteristic? Therefore, this paper chooses to examine the spatial decay under a geographic distance matrix, drawing on the measurement method by Feng et al. [27], and considering the distances between the port cities under study. The measurement starts from 55km, setting a threshold every 5km (55, 60, 80, 85, 90, 95, 100, 105, 110). Subsequently, a local spatial weight matrix is reconstructed. It should be noted that since the decay matrices corresponding to thresholds at 65, 70, and 75km are the same as the decay matrix for the 60km threshold, matrices for d = 65, 70, and 75 were not set in this paper.

Table 7 shows the direct effects, indirect effects (spillover effects), and total effects of ports on urban economies at different distance thresholds. From the lnPT_Indirect column, we can observe the attenuation of the port's impact on urban economies; at the 55km threshold, the spillover effect decreases from 1.6733 to 0.5018, showing a significant attenuation. Within the range of 55km to 105km, the spillover effects of ports on urban economies fluctuate and decline, particularly noticeable rebounds at 85km and 100km, but a significant decline occurs between 100km and 105km. By the time it reaches 110km, the spillover effect of ports on urban economies no longer shows statistical significance, thus attenuating to zero. Overall, the spillover effect exhibits a fluctuating downward trend.

To facilitate a more intuitive understanding, this paper has created scatter plots and locally weighted regression smoothing graphs to illustrate the attenuation of the port's economic spill-over effects on surrounding cities as distance increases. Fig 1 displays the relationship between Indirect (the economic spillover effect on urban economies) and Distance (the distance

**Table 7. Regression results of decay trend.**

| d | lnPT_Direct | lnPT_Indirect | lnPT_Total | ρ | Log-Likelihood |
|---|---|---|---|---|---|
| d = 55 | 0.0695*** | 0.5108*** | 0.5803*** | 0.5364*** | 399.7926 |
|  | (3.68) | (3.16) | (3.33) | (5.38) |  |
| d = 60 | 0.0754*** | 0.5420*** | 0.6174*** | 0.4124*** | 378.3173 |
|  | (3.79) | (3.52) | (3.75) | (3.40) |  |
| d = 80 | 0.0689*** | 0.5288*** | 0.5976*** | 0.3939*** | 377.9466 |
|  | (3.49) | (3.57) | (3.76) | (3.16) |  |
| d = 85 | 0.0726*** | 0.6948*** | 0.7675*** | 0.3656*** | 385.3825 |
|  | (3.94) | (4.28) | (4.46) | (2.89) |  |
| d = 90 | 0.0722*** | 0.5023*** | 0.5745*** | 0.2674* | 369.9624 |
|  | (3.81) | (3.91) | (4.17) | (1.90) |  |
| d = 95 | 0.0719*** | 0.5295*** | 0.6014*** | 0.3120** | 367.4026 |
|  | (3.67) | (3.56) | (3.81) | (2.27) |  |
| d = 100 | 0.0738*** | 0.6425*** | 0.7163*** | 0.3557*** | 374.7159 |
|  | (3.71) | (4.00) | (4.16) | (2.70) |  |
| d = 105 | 0.0665*** | 0.2176* | 0.2841** | 0.3134** | 363.5414 |
|  | (3.48) | (1.69) | (2.08) | (2.35) |  |
| d = 110 | 0.0511*** | 0.0607 | 0.1118 | 0.2330* | 360.2747 |
|  | (2.58) | (0.52) | (0.91) | (1.67) |  |

between cities). It is apparent that as distance increases, the economic spillover effect of ports on urban economies shows a fluctuating downward trend, tending towards zero at 110km. In summary, these results corroborate the validity of Hypothesis 4.

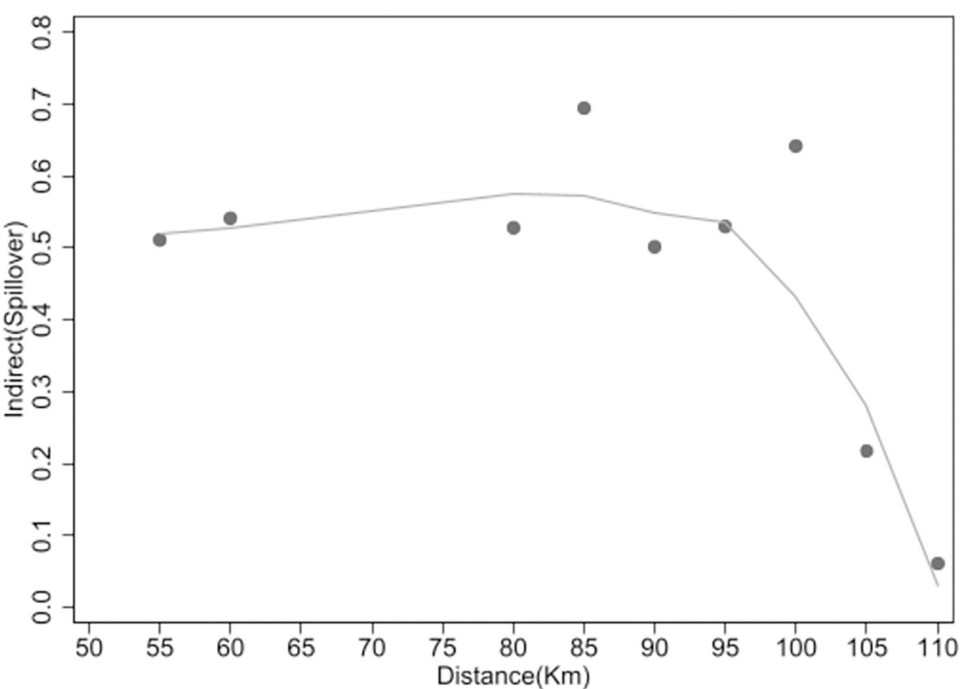

**Fig 1. Decay trend.**

## 5.3. The role of financial development in the impact of ports on regional economies

Does the effect of ports on urban economic growth vary with financial development? Based on this, this paper constructed a financial development distance matrix to preliminarily investigate the role of financial development in the effect of ports on urban economic growth, with results presented in Table 8. Under the financial development distance matrix, whether it be local effects, neighboring (spillover) effects, or total effects, all are significant at the 1% significance level, indicating that with the support of financial development, ports can not only promote local urban economic growth but also radiate to neighboring areas, boosting their economic growth. Therefore, we have reason to believe that financial development plays a threshold effect in the impact of ports on urban economic growth, which also verifies the first part of Hypothesis 5.

Table 9 employs the "Bootstrap" method to calculate the dual-threshold effect of financial development in the impact of ports on urban economic growth and the corresponding values. The results show a dual-threshold effect of financial development. The first threshold value is 17.1233, significant at the 1% level, and the second threshold value is 17.9721, also significant at the 1% level. Hence, we can conclude that financial development has a nonlinear impact on the role of ports in urban economic growth, that is, different levels of financial development affect the degree of contribution of ports to urban economic growth. This validates the latter part of Hypothesis 5.

Through Table 8, we can gain a detailed understanding of this nonlinear impact. When financial development is below the first threshold value of 17.1233, the impact coefficient of ports on urban economic growth is 0.1017, significant at the 1% level. This indicates that at lower levels of financial development, ports play a role in driving urban economic growth. When financial development exceeds the first threshold value of 17.1233 but is below the second threshold value of 17.9721, its impact coefficient is 0.1308, significant at the 1% level. This

**Table 8. The impact of financial development.**

| Variable | Direct | Indirect | Total | FIN | Value |
|---|---|---|---|---|---|
| lnPT | 0.0959*** | 0.2676*** | 0.3994*** | lnPT (I≤17.1233) | 0.1017** |
| | (5.30) | (4.82) | (5.24) | | (2.51) |
| lnSSL | -0.1556*** | -0.2508*** | -0.5892*** | lnPT (17.12233< I ≤17.9721) | 0.1308*** |
| | (-3.64) | (-2.97) | (-4.82) | | (3.25) |
| lnCSL | -0.0869 | 0.9813*** | 0.3561 | lnPT (I>17.9721) | 0.1542*** |
| | (-0.73) | (3.42) | (0.83) | | (3.82) |
| INF | -0.4606 | -9.6992*** | -7.5894 | lnSSL | 0.5921*** |
| | (-0.54) | (-2.64) | (-1.63) | | (9.30) |
| lnHUM | -0.1076** | 0.0108 | -0.0444 | lnCSL | 0.3894 |
| | (-2.26) | (0.09) | (-0.24) | | (1.53) |
| ρ | 0.3047*** | | | INF | 1.9224 |
| | (3.15) | | | | (0.99) |
| Observations | 200 | | | lnHUM | 0.4364*** |
| | | | | | (4.28) |
| Log-likelihood | 374.0166 | | | Constant | 2.7482*** |
| | | | | | (5.33) |
| Number of ID | 20 | | | Observations | 200 |
| - | - | | | R-squared | 0.842 |
| - | - | | | Number of ID | 20 |

**Table 9. Threshold effect analysis.**

| Variable | Number | Threshold Value | P-Value | F-Value | Critical Value | | |
|---|---|---|---|---|---|---|---|
| | | | | | 1% | 5% | 10% |
| FIN | Single | 17.1233 | 0.0000 | 89.14*** | 54.4097 | 39.1774 | 34.1975 |
| | Double | 17.9721 | 0.0000 | 61.40*** | 38.3805 | 30.5654 | 25.4882 |

suggests that at higher levels of financial development, this driving effect is enhanced. When financial development is above the second threshold value of 17.9721, its impact coefficient is 0.1542, significant at the 1% level. Although the enhancement effect is smaller, the driving effect of financial development continues. In summary, the latter part of Hypothesis 5 is validated.

# 6. Heterogeneity

## 6.1. Port heterogeneity

This article, based on the classification in the "China Port Statistical Yearbook," divides the ports of the YRD into coastal ports and inland ports. Here, we refer to coastal ports as CPC and inland ports as IPC, with specific division information in Table 10. The corresponding IDs 1–20 are Shanghai, Nanjing, Wuxi, Xuzhou, Changzhou, Suzhou, Nantong, Lianyungang, Huaian, Yancheng, Yangzhou, Zhenjiang, Taizhou, Hangzhou, Ningbo-Zhoushan, Jiaxing, Huzhou, Shaoxing, Wenzhou, Taizhou, respectively.

From the results in Table 11, it is evident that the spatial autoregressive coefficients ρ for both coastal and inland ports are significant at the 10% significance level, indicating that there is still a spatial dependency among urban economies within the YRD ports when divided into types of ports. Simultaneously, the direct effects of both coastal and inland ports are significant at the 5% significance level, suggesting that both coastal and inland ports can promote local urban economic growth. However, regarding the spillover effect on urban economies, coastal ports have a spillover effect of 0.1128, but it no longer shows statistical significance, thus indicating a lack of spillover effect. In contrast, the spillover effect of inland ports on urban economies is 0.365, which is significant at the 1% significance level. This indicates that the development of inland ports has a spillover effect, can radiate to the surrounding areas, and promote economic growth in neighboring cities. Moreover, observing the spillover effects, inland ports at 0.3650 exceed coastal ports at 0.1128, suggesting that inland ports have a greater advantage in generating spillover effects on the economies of surrounding areas. Synthesizing the above analysis, while coastal port development does not exhibit spillover effects on

**Table 10. Port heterogeneity.**

| ID | 1 | 2 | 3 | 4 | 5 | 6 | 7 |
|---|---|---|---|---|---|---|---|
| Indirect | 0.2048 | 0.3536 | 0.2092 | 1.4953*** | 0.2690 | 1.1804*** | 1.1126** |
| Port | CPC | IPC | IPC | IPC | IPC | IPC | IPC |
| ID | 8 | 9 | 10 | 11 | 12 | 13 | 14 |
| Indirect | 0.4349*** | 0.6997 | -0.0417 | 0.1566 | -0.2281 | 0.6725*** | 0.1851 |
| Port | CPC | IPC | CPC | IPC | IPC | IPC | IPC |
| ID | 15 | 16 | 17 | 18 | 19 | 20 | |
| Indirect | 0.6250 | -1.5834*** | 0.5883* | 0.1291 | 0.2736 | 0.9177 | |
| Port | CPC | CPC | IPC | IPC | CPC | CPC | |

**Table 11. Port-specific regression.**

| Variable | CPC | | IPC | |
|---|---|---|---|---|
| | Direct | Indirect | Direct | Indirect |
| lnPT | 0.0822** | 0.1128 | 0.0838*** | 0.3650*** |
| | (2.00) | (1.59) | (4.08) | (4.57) |
| lnSSL | -0.0939 | 0.0190 | -0.1286*** | -0.6111*** |
| | (-1.32) | (0.16) | (-2.73) | (-4.73) |
| lnCSL | -0.3162 | -1.3173** | -0.0425 | -0.3670 |
| | (-1.17) | (-2.38) | (-0.34) | (-0.68) |
| INF | -0.6003 | -1.3253 | -1.3637 | -3.5247 |
| | (-0.36) | (-0.40) | (-0.73) | (-0.54) |
| lnHUM | 0.1184 | 0.2209 | 0.0096 | 0.7421*** |
| | (1.36) | (1.43) | (0.18) | (3.66) |
| $\rho$ | 0.4077*** | | 0.2646* | |
| | (3.11) | | (1.77) | |
| Observations | 70 | | 130 | |
| Log-likelihood | 146.7629 | | 267.5511 | |
| Number of ID | 7 | | 13 | |

neighboring economic growth, inland ports demonstrate a positive spillover effect, thereby partially validating Hypothesis 3.

## 6.2. Urban heterogeneity

This paper draws on the research method of Aquaro et al. [33], constructing a Spatial Durbin Model with heterogeneous coefficients to study the spillover effect of each port on the economic growth of neighboring urban areas. The indicators in formula (8) have the same meaning as in formula (2).

$$\ln PGDP = \alpha_0 + \rho_i W \cdot \ln PGDP + \alpha_{1i} \ln PT_{it} + \beta_{1i} W \cdot \ln PT_{it}$$

$$+ \alpha_{ki} X_{it} + \beta_{ki} W \cdot X_{it} + \mu_i + \lambda_t + \varepsilon_{it} \tag{8}$$

Table 10 reveals that among all the ports studied, 7 ports have significant coefficients for their spatial lag terms, with Xuzhou, Suzhou, Nantong, Lianyungang, Taizhou, and Huzhou showing positive significance. This indicates that the ports in these areas exert a positive impact on the economies of neighboring cities. Further analysis indicates that Xuzhou, Suzhou, Nantong, Taizhou, and Huzhou are inland ports, further underscoring the advantage of inland ports in generating positive spillover effects on nearby urban economies. Conversely, Jiaxing, as a coastal port, has a significant negative coefficient, suggesting a "siphoning effect" on the economies of surrounding cities.

## 7. Robustness and endogeneity

### 7.1. Replacing weight matrices

Table 12 showcases the impact of ports on urban economic growth and their spillover effects under two different spatial weight matrices to verify the robustness of the conclusions. Across all spatial weight matrices, the spatial autocorrelation coefficient $\rho$ is significantly positive at the 1% level, confirming spatial correlations between ports and urban economic growth. Moreover, both local effects and neighboring effects of ports on urban economic growth pass

**Table 12. Replacing weight matrices.**

| Variable | W$_{0-1}$ | | W$_2$ | |
|---|---|---|---|---|
| | Direct | Indirect | Direct | Indirect |
| lnPT | 0.0615*** | 0.2950*** | 0.0934*** | 0.7131*** |
| | (3.56) | (5.23) | (4.75) | (4.05) |
| lnSSL | -0.1077*** | -0.5552*** | -0.1563*** | -1.1718*** |
| | (-3.08) | (-6.27) | (-4.56) | (-4.80) |
| lnCSL | 0.0832 | 0.4898 | -0.0504 | -0.3176 |
| | (0.70) | (1.43) | (-0.42) | (-0.36) |
| INF | -2.2265** | -5.1985* | -1.9377* | -14.3593 |
| | (-2.39) | (-1.75) | (-1.92) | (-1.59) |
| lnHUM | -0.0612 | -0.3142** | -0.0567 | -0.2133 |
| | (-1.32) | (-2.38) | (-1.19) | (-0.63) |
| ρ | 0.4346*** | | 0.5358*** | |
| | (6.38) | | (5.46) | |
| Observations | 200 | | 200 | |
| Log-likelihood | 394.4462 | | 403.5533 | |
| Number of ID | 20 | | 20 | |

the 1% significance level test, with all effect coefficients being positive. Furthermore, across all weight matrices, the spillover effects exceed direct effects, indicating that ports play a more advantageous role in promoting economic growth in surrounding areas than in local economies. Thus, the conclusions of this paper are robust.

## 7.2. Replacing the dependent variable

This study includes urban GDP as the dependent variable in the model for re-regression. The column lnGPD in Table 13 indicates that after incorporating urban GDP into the model, the spatial autocorrelation coefficient ρ is significant at the 5% level, suggesting significant spatial

**Table 13. Replacing the dependent variable.**

| lnGDP | | | Variable | (1) Value | Variable | (2) Value |
|---|---|---|---|---|---|---|
| Variable | Direct | Indirect | | | | |
| lnPT | 0.0352** | 0.4241*** | L. lnPT | 0.1011*** | L2. lnPT | 0.0805** |
| | (2.08) | (4.09) | | (3.44) | | (2.56) |
| lnSSL | 0.0739** | -0.6596*** | lnSSL | -0.0065 | lnSSL | -0.0356 |
| | (2.21) | (-4.71) | | (-0.10) | | (-0.59) |
| lnCSL | -0.2127** | 0.4381 | lnCSL | 0.0024 | lnCSL | -0.4322 |
| | (-2.00) | (0.76) | | (0.01) | | (-1.19) |
| INF | 1.0780 | 4.8324 | INF | 0.3163 | INF | 2.5826 |
| | (1.28) | (0.79) | | (0.09) | | (0.89) |
| lnHUM | -0.0277 | -0.0658 | lnHUM | 0.1800*** | lnHUM | 0.1950*** |
| | (-0.64) | (-0.29) | | (4.89) | | (5.70) |
| ρ | 0.2931** | | _cons | 10.0414*** | _cons | 10.6648*** |
| | (2.33) | | | (17.87) | | (19.95) |
| Observations | 200 | | 180 | | 160 | |
| Log-likelihood | 402.9615 | | Kleibergen-Paap rk LM | | 49.536*** | |
| Number of ID | 20 | | Kleibergen-Paap rk Wald F | | 1620.217*** | |

correlation between ports and urban economies. Further analysis of the direct and indirect effects of ports reveals that their coefficients are 0.0352 and 0.4241, respectively (with spillover effects exceeding direct effects), both significant at the 5% level. This implies that ports can promote local urban economic growth as well as that of neighboring areas, proving the robustness of the conclusions of this paper.

### 7.3. Two-stage least squares regression

Through the analysis above, this paper employs a SDM with both individual and time fixed effects to minimize measurement errors. However, there may exist issues of endogeneity within the model construction process. To circumvent such issues, this study adopts the two-stage least squares method, incorporating the port development level lagged by one period (L. lnPT) and two periods (L2. lnPT) as instrumental variables for regressing with urban economic growth. As shown in columns (1) and (2) of Table 13, even after accounting for endogeneity, ports continue to exert a significant influence on urban economic growth, underscoring the robustness of our findings. Moreover, the P-values corresponding to the Kleibergen-Paap rk LM and Kleibergen-Paap rk Wald F statistics are zero, indicating the successful exclusion of under-identification and weak instrumental variables issues. Finally, examining the endogeneity of variables via the Hausman test yielded a P-value of 0.677, accepting the null hypothesis that all variables are exogenous.

## 8. Discussion

The YRD region boasts a significantly higher number of international hub ports compared to other regions, offering unique advantages for rapid economic development and regional integration. This raises the question of the impact these ports have on the regional economy and to what extent. With these considerations in mind, this paper aims to employ an SDM to investigate the impact and spillover effects of ports on regional economies in this area. However, upon reviewing related literature, the study by Deng et al. (2020) was found to be the most similar [16], in which they calculated the composite strength of China's five major coastal port clusters and explored how the comprehensive strength of ports affects the economic growth of local and surrounding cities, including the YRD region. A closer examination of this article reveals some differences.

The differences are mainly in two aspects: (1) The article did not consider how the distance between cities affects the spillover effect of ports. Essentially, as the distance for transporting products from ports to surrounding areas increases, logistic costs rise, reducing the competitiveness of products in the receiving areas. This leads to a decline in profits and economic growth, thereby weakening the positive radiative effect of ports. Therefore, neglecting distance in studying spillover effects may introduce bias and affect related conclusions. (2) The article also analyzed how urban capital impacts the economy of coastal port areas, noting specifically that an increase in urban capital has a significant direct impact on local economic growth and has a positive spatial spillover effect on the economic growth of surrounding cities. However, this differs from the content this paper intends to study, which considers the role of ports in influencing urban economic growth, as well as the effects of financial development, rather than conducting a spatial regression between financial development and urban economic growth, which overlooks the role of ports. Thus, this paper considers combining the three for analysis. On one hand, this paper establishes a financial development distance weight matrix, taking it into account when analyzing the spillover effect of ports on urban economic growth. On the other hand, this paper uses a threshold effect model to study the role financial development plays at different levels in the impact of ports on urban economic growth.

Reviewing our article reveals two conclusions not previously addressed by prior research. (1) Compared to the direct driving effect of ports on the economic growth of their local cities, the indirect driving effect of ports on the economic growth of surrounding cities is more pronounced. This conclusion was drawn by comparing the direct and indirect effects of ports, and it remains valid under related robustness tests. The economic growth of a city where a port is located is not solely dependent on the port as a growth pole; other growth poles may be more significant. Additionally, this paper selects port throughput as the indicator of port development level, which is overly simplistic and fails to consider the comprehensive impact generated by ports. Ports can produce a significant economic radiative force and driving effect through multiple pathways, such as promoting regional economic integration, industrial diffusion, infrastructure improvement, job creation, attracting external investment, and enhancing competitiveness and openness. Therefore, focusing solely on the role of ports might underestimate their impact on the local economy. Based on these findings, the areas for improvement and future research directions identified in this paper are to find suitable and compelling evaluation indicators to more accurately assess the impact of ports on urban economic growth and spillover effects.

(2) Compared to coastal ports, which do not exhibit a spillover effect on the economies of surrounding cities, inland ports demonstrate a positive spillover effect. This conclusion was derived from dividing the YRD port cluster into coastal and inland ports and conducting group regression analysis. The difference arises because coastal ports have a unique advantage in international trade, with their economic activities spread globally, potentially resulting in weaker economic ties with surrounding regions compared to inland ports. However, extending the perspective globally to study the spillover effects of coastal ports on other countries might yield different conclusions. Since China's reform and opening, coastal port cities in the YRD have developed rapidly due to their geographical advantages (not implying that all coastal cities are port cities). This has led to highly uneven development among coastal port cities, which could impact the conclusions of this paper. As discussed in the mechanism analysis section, inland ports are more involved in domestic trade and the domestic economic cycle, entailing more domestic distribution and consumption, thus having a comparative advantage in strengthening economic ties with domestic surrounding areas. Moreover, compared to the economic disparities among coastal port cities, the economic development gap among inland port cities is smaller and relatively balanced. This also influences the conclusions of this paper. Based on this analysis, the areas for improvement and future research directions involve finding suitable samples to intricately study the differences and underlying reasons between coastal and inland port cities in driving the economy of surrounding domestic regions.

## 9. Conclusions and policy recommendations

To explore the impact of the YRD ports on the region's economy, this paper first conducts basic regression with port throughput, the distance of each city from Shanghai, and the level of urban economic development. This demonstrates that port throughput influences urban economic development and that there exists a spatial connection between the distance of cities and urban economic development. In the second step, spatial econometric testing methods are employed to test if the data can analyze the spatial lag of independent variables using the spatial Durbin model. This proves the rationality of the chosen model and that ports have a positive impact on the local and surrounding regional economy. It also finds that, compared to the direct driving effect on local urban economic growth, the YRD ports have a more significant indirect driving effect on the economic growth of surrounding cities, like the study conducted by Munim and Schramm [22]. The third step examines how different distances between cities

affect the economic spillover effect of ports, finding that the positive spillover effect of ports fluctuates mildly between 55-95km and then drops sharply after 95km until it disappears at 110km, aligning with the findings of Yudhistira and Sofiyandi study [34]. The fourth step constructs a financial development distance matrix and threshold effect model, incorporating the impact of financial development in assessing the influence of ports on urban economies. It finds that financial development has a positive driving effect with a clear threshold effect, Similar to the study conducted by Hesse and Rodrigue [18]. Finally, by dividing the YRD ports into coastal and inland ports and conducting spatial regression in groups, it is found that they exhibit different spillover effects on surrounding areas, with inland ports showing a positive spillover and coastal ports showing none, aligning with the findings of Han et al. [23]. Further findings indicate that individual ports have varying spillover effects on their surroundings, but those with positive spillover effects are mostly inland ports. Based on these conclusions, this paper suggests the following:

(1) Fully leverage the local role of ports in promoting urban economic growth. On one hand, as gateways to international trade, ports should actively expand into overseas markets and attract foreign investment to foster industrial upgrading and economic diversification in cities. This includes establishing free trade zones and economic special zones, offering tax incentives and administrative conveniences to attract high-value industries such as financial services, electronic information, and biotechnology to cluster, enhancing the city's international competitiveness, promoting employment and innovation, and driving overall urban economic growth. On the other hand, ports should enhance interconnectivity with the city's inland transportation network to improve logistics efficiency and reduce transportation costs. Investing in efficient logistics systems, such as intelligent transportation management systems, and improving the quality of road and rail connections are essential to ensure goods can be rapidly and safely transferred from the port to various inland destinations. Moreover, by cooperating with local universities and research institutions, ports can become centers for innovation and technological development, guiding the application and promotion of new technologies and business models within the city, further stimulating economic vitality and sustained growth.

(2) Strengthen the port's radiating effect on the economic growth of neighboring cities. Firstly, the government should enhance the connectivity of roads, railways, and waterways to modernize the transport logistics network, actively strengthening the connection between ports and surrounding cities. Encourage local ports and adjacent areas to collaborate deeply to maximize the spillover effect of ports on the economic development of neighboring regions. Secondly, border effects and geographical decay make it difficult for areas beyond the decay boundary to enjoy the developmental dividends of port cities. On one hand, different cities can cooperate to build a regional information sharing platform to accelerate the construction of data standardization, ensuring data can correctly be exchanged and shared across various systems and applications, thus slowing the speed of geographical decay. On the other hand, the government actively supports the integration of the YRD, weakens local protectionism, cooperates actively with surrounding areas to promote the free flow of information between regions, thereby expanding the spatial scope of positive spillover effects and extending the decay boundary.

(3) Create a macroeconomic environment conducive to capital flow and investment. Formulate a set of strategies to encourage and guide financial capital towards investment in port infrastructure and related industries. These strategies include tax incentives, deepening reforms of the financial market, and support for financial product innovation. Additionally, pay attention to the regional layout and functional expansion of financial services to ensure sufficient funding to support the sustained development of ports and their surrounding areas.

Meanwhile, establish risk management mechanisms to balance the returns and risks of capital investment, maintaining stable economic growth in port cities. Through these comprehensive measures, enhance the financial vitality of port cities, safeguarding the continued prosperity of the economy.

Governments of cities in the YRD where ports are located should, under the guidance of the national integration development plan for the YRD, consider the macroeconomic situation and the economic development of surrounding cities. They should strive to create core areas with diffusion and driving effects, enhancing the positive radiation of developed cities to less developed ones, and promoting economic and trade development in surrounding areas. Economically less developed cities should fully leverage the spatial spillover effects generated by the higher development level of nearby ports to compensate for their own economic development or shortage of production factor resources, ultimately achieving shared prosperity in the YRD region.

## Supporting information

**S1 Data.**
(XLSX)

## Acknowledgments

We would like to thank the reviewers for providing professional comments on the manuscript.

## Author Contributions

**Conceptualization:** Liangyu Chen.

**Funding acquisition:** Jian Hou.

**Methodology:** Liangyu Chen.

**Software:** Zhouping Zhang.

**Supervision:** Jian Hou.

**Visualization:** Zhouping Zhang.

**Writing – original draft:** Jian Hou, Edwin Kuang.

**Writing – review & editing:** Juming Shi.

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
