## [Decision Letter · Decision Letter 0]

22 Feb 2024

PONE-D-23-38795Exploring the spatial spillover implications of ports in the Yangtze River Delta region on the economic prosperity of citiesPLOS ONE

Dear Dr. Shi,

Thank you for submitting your manuscript to PLOS ONE. After careful consideration, we feel that it has merit but does not fully meet PLOS ONE’s publication criteria as it currently stands. Therefore, we invite you to submit a revised version of the manuscript that addresses the points raised during the review process.

Authors need to make their research more presentable. Please go over the suggestions made by the reviewers carefully especially Reviewer 1** **==============================

We look forward to receiving your revised manuscript.

Kind regards,

Muhammad Khalid Bashir, PhD

Academic Editor

PLOS ONE

 [National Social Science Fund Project: Research on the Mechanism of Adequate Dimensionality Reduction in Complex Data and Its Application in the Field of Ecological Environment (Project No. 22BTJ018).].  

5. In the online submission form, you indicated that [The data underlying the results presented in the study are available from corresponding authors.]. 

Reviewers' comments:

Reviewer's Responses to Questions

**Comments to the Author**

1. Is the manuscript technically sound, and do the data support the conclusions?

Reviewer #1: Yes

Reviewer #2: Yes

2. Has the statistical analysis been performed appropriately and rigorously? 

Reviewer #1: Yes

Reviewer #2: Yes

3. Have the authors made all data underlying the findings in their manuscript fully available?

Reviewer #1: Yes

Reviewer #2: Yes

4. Is the manuscript presented in an intelligible fashion and written in standard English?

Reviewer #1: No

Reviewer #2: No

5. Review Comments to the Author

Reviewer #1: This paper selects panel data from 20 ports in the Yangtze River Delta region and their host cities, then carries out the empirical examination of the implications that ports have on the prosperity of urban economies. The topic is worth studying, but the article still has some problems that need to be corrected:

1. A literature review was conducted on “the interrelation between port enhancement and regional economic prosperity” and “the interrelation between financial development and economic prosperity”, respectively, but should logically link them together. Meanwhile, “literature review” is not simply a list of literature.

2. The theoretical value and practical implications of the study are not clearly presented in the introduction or literature review.

3. Instead of listing each common models, the research article needs to articulate the applicability and strengths of the chosen one.

4. What is the reason behind the conclusion of the paper that “the indirect stimulus on the economies of surrounding cities is more pronounced compared to the direct implications on the economies of the host cities”?

5. Generally, research articles need to have an “Discussions” section before “Conclusions”, in which they need to go deeper and improve on the previous research. I suggest that you should restructure this article.

6. The writing of this article is not standard, with many grammatical and formatting errors, such as incorrect line numbers. I strongly advise seeking the assistance of a native English speaker or a professional language editor to enhance the clarity and coherence of your paper.

Reviewer #2: I thoroughly reviewed the manuscript title “Exploring the spatial spillover implications of ports in the Yangtze River Delta region on the economic prosperity of cities”. Based on several reasons I would recommend major revisions of the manuscript.

Some of my points are as under:

The language and sentence structures of this manuscript are at times incomprehensible. The writing style in general is awkward at too many instances. The paper needs extensive editing.

This is basically repetition of existing work. Similar studies have also been published in other journals. Author(s) should add a separate paragraph (in the Introduction Section) explaining the shortcomings of previous research and how this study is different from existing studies. I mean contribution of this study in existing literature.

I found several inconsistencies in the 2nd section “Literature review and theoretical hypotheses”.

Author(s) should discuss/explain hypotheses in the last paragraph of the Introduction Section.

Rewrite hypotheses in a scientific way.

Similarly, writing style in the Literature Review section is awkward. Author(s) should add a separate section only on literature review: provide the comments of the cited papers after introducing each relevant work. What readers require is, by convinced literature review, to understand the clear thinking/consideration why the proposed approach can reach more convinced results. This is the very contribution from authors. In addition, authors also should provide more sufficient critical literature review to indicate the drawbacks of existed approaches, then, well define the main stream of research direction, how did those previous studies perform? Employ which methodologies? Which problem still requires to be solved? Why is the proposed approach suitable to be used to solve the critical problem? We need more convinced literature reviews to indicate clearly the state-of-the-art development.

The models used in the analyses are appropriate.

The Discussion Section is missing. Author(s) should add the Discussion Section and that must have a convincing and strong literature to support the findings and unfortunately the writing style in general in the results section is awkward.

The conclusion needs revision. Authors should write what they have actually concluded based on the study’s findings. It should not be the repetition of the abstract.

All the best!

6. PLOS authors have the option to publish the peer review history of their article (what does this mean?). If published, this will include your full peer review and any attached files.

Reviewer #1: No

Reviewer #2: No

---

## [Author Response · Author response to Decision Letter 0]

30 Mar 2024

Thank you very much for your comments and suggestions. In the following, we copy your comments in italic and follow with our response. We kindly request that reviewers consider our page numbers and line numbers as the reference due to potential inconsistencies with the system's format. We deeply appreciate your understanding of this matter. 

The related details are provided in the "Response to reviewers" document.

---

## [Decision Letter · Decision Letter 1]

11 Jul 2024

Exploring the spatial spillover effects of Yangtze River Delta ports on urban economic growth

PONE-D-23-38795R1

Dear Dr. Shi,

We’re pleased to inform you that your manuscript has been judged scientifically suitable for publication and will be formally accepted for publication once it meets all outstanding technical requirements.

Kind regards,

Muhammad Khalid Bashir, PhD

Academic Editor

PLOS ONE

Additional Editor Comments (optional):

Authors are requested to re-write the Abstract in order to reflect conclusions and implications.

Reviewers' comments:

Reviewer's Responses to Questions

**Comments to the Author**

1. If the authors have adequately addressed your comments raised in a previous round of review and you feel that this manuscript is now acceptable for publication, you may indicate that here to bypass the “Comments to the Author” section, enter your conflict of interest statement in the “Confidential to Editor” section, and submit your "Accept" recommendation.

Reviewer #2: All comments have been addressed

Reviewer #3: All comments have been addressed

2. Is the manuscript technically sound, and do the data support the conclusions?

Reviewer #2: Yes

Reviewer #3: Yes

3. Has the statistical analysis been performed appropriately and rigorously? 

Reviewer #2: Yes

Reviewer #3: Yes

4. Have the authors made all data underlying the findings in their manuscript fully available?

Reviewer #2: Yes

Reviewer #3: Yes

5. Is the manuscript presented in an intelligible fashion and written in standard English?

Reviewer #2: Yes

Reviewer #3: Yes

6. Review Comments to the Author

Reviewer #2: I have thoroughly reviewed the revision of the manuscript. The paper can be accepted for publication.

Reviewer #3: This articel is designed to explore the spatial spillover effects of Yangtze River Delta ports on urban economic growth. The authors found that: (1) The YRD ports significantly contribute to economic growth in both the port cities and their surrounding areas, with the indirect impact on neighboring cities being more substantial than the direct effect on the cities themselves; (2) The beneficial spillover effects of the YRD ports on the economic growth of nearby cities vary in intensity over different spatial ranges, marked by distinct boundary effects and geographical attenuation. The influence extends up to approximately 110km; (3) Within the various elements impacting the economic growth of cities in the YRD, financial development prominently exhibits a threshold effect on urban economic growth; (4) Upon analyzing heterogeneity, inland and coastal port cities manifest divergent spillover effects, with inland port cities predominantly exerting a positive spillover on adjacent regions.. The overall methodology and procedures were valid, and it's well written. However, the ABSTRACT should be re-summarized as there is no conclusions and implications.

7. PLOS authors have the option to publish the peer review history of their article (what does this mean?). If published, this will include your full peer review and any attached files.

Reviewer #2: No

Reviewer #3: No

---

## [Editor Report · Acceptance letter]

7 Aug 2024

PONE-D-23-38795R1 

PLOS ONE

Dear Dr. Shi, 

I'm pleased to inform you that your manuscript has been deemed suitable for publication in PLOS ONE. Congratulations! Your manuscript is now being handed over to our production team.

Kind regards, 

on behalf of

Dr. Muhammad Khalid Bashir 

Academic Editor

PLOS ONE